# Multi-Source Diffusion Models for Simultaneous Music Generation and Separation

**Giorgio Mariani**[*]
Sapienza University of Rome
`mariani@di.uniroma1.it`

**Irene Tallini**[*]
Sapienza University of Rome
`tallini@di.uniroma1.it`

**Emilian Postolache**[*]
Sapienza University of Rome
`postolache@di.uniroma1.it`

**Michele Mancusi**[*]
Sapienza University of Rome
`mancusi@di.uniroma1.it`

**Luca Cosmo**[†]
Ca' Foscari University of Venice
`luca.cosmo@unive.it`

**Emanuele Rodolà**[†]
Sapienza University of Rome
`rodola@di.uniroma1.it`

## Abstract

In this work, we define a diffusion-based generative model capable of both music synthesis and source separation by learning the score of the joint probability density of sources sharing a context. Alongside the classic total inference tasks (i.e., generating a mixture, separating the sources), we also introduce and experiment on the partial generation task of source imputation, where we generate a subset of the sources given the others (e.g., play a piano track that goes well with the drums). Additionally, we introduce a novel inference method for the separation task based on Dirac likelihood functions. We train our model on Slakh2100, a standard dataset for musical source separation, provide qualitative results in the generation settings, and showcase competitive quantitative results in the source separation setting. Our method is the first example of a single model that can handle both generation and separation tasks, thus representing a step toward general audio models.

## 1 Introduction

Generative models have recently gained much attention thanks to their successful application in many fields, such as NLP (OpenAI, 2023; Touvron et al., 2023; Santilli et al., 2023), image synthesis (Ramesh et al., 2022; Rombach et al., 2022) or protein design (Shin et al., 2021; Weiss et al., 2023; Minello et al., 2024). Audio is no exception to this trend (Agostinelli et al., 2023; Liu et al., 2023).

A peculiarity of the audio domain is that an audio sample $\mathbf{y}$ can be seen as the sum of multiple individual sources $\{\mathbf{x}_1, \ldots, \mathbf{x}_N\}$, resulting in a mixture $\mathbf{y} = \sum_{n=1}^{N} \mathbf{x}_n$. Unlike in other sub-fields of the audio domain (e.g., speech), sources present in musical mixtures (stems) share a *context* given their strong interdependence. For example, the bass line of a song follows the drum's rhythm and

---

[*]Equal contribution. Listing order is random. G.M. wrote most of the code, performed most objective experiments, and contributed to the development of the Dirac separator. I.T. proposed and developed the idea of the Dirac separator and partly formalized its proof, contributed to the code and the objective experiments, especially concerning the Dirac separator, performed the subjective listening tests, and wrote substantial parts of the paper. E.P. proposed and developed the ideas of the source-joint Bayesian separator and the sub-FAD metric, partly formalized the proof of the Dirac separator, contributed to the code and the objective experiments, especially concerning source imputation, and wrote substantial parts of the paper. M.M. proposed the idea of using the source-joint model for music (and accompaniment) generation, proposed using the correction steps, and contributed to the objective experiments.

[†]Shared last authorship.

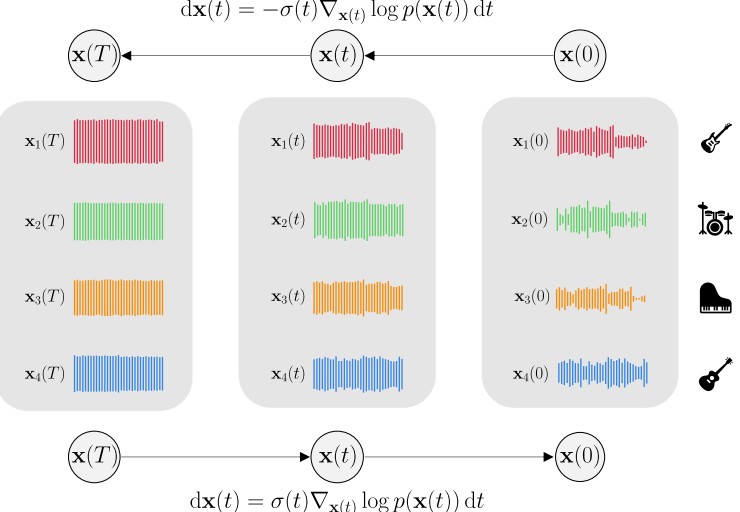

**Figure 1: Proposed method.** We leverage a forward Gaussian process (right-to-left) to learn the score over contextual sets (the large boxes) of instrumental sources (the waveforms) across different time steps $t$. During inference, the process is reversed (left-to-right), letting us perform the tasks of total generation, partial generation, or source separation (Figure 2).

harmonizes with the melody of the guitar. Mathematically, this fact can be expressed by saying that the joint distribution of the sources $p(\mathbf{x}_1, \ldots, \mathbf{x}_N)$ does *not* factorize into the product of individual source distributions $\{p_n(\mathbf{x}_n)\}_{n=1,\ldots,N}$. Knowing the joint $p(\mathbf{x}_1, \ldots, \mathbf{x}_N)$ implies knowing the distribution over the mixtures $p(\mathbf{y})$ since the latter can be obtained through the sum. The converse is more difficult mathematically, being an inverse problem.

Nevertheless, humans have developed the ability to process multiple sound sources simultaneously in terms of synthesis (i.e., musical composition or generation) and analysis (i.e., source separation). More specifically, composers can invent multiple sources $\{\mathbf{x}_1, \ldots, \mathbf{x}_N\}$ that sum to a consistent mixture $\mathbf{y}$ and, extract information about the individual sources $\{\mathbf{x}_1, \ldots, \mathbf{x}_N\}$ from a mixture $\mathbf{y}$.

This ability to compose and decompose sound is crucial for a generative music model. A model designed to assist in music composition should be capable of isolating individual sources within a mixture and allow for independent operation on each source. Such a capability would give the composer maximum control over what to modify and retain in a composition. Therefore, we argue that compositional (waveform) music generation is highly connected to music source separation.

To our knowledge, no model in deep learning literature can perform both tasks simultaneously. Models designed for the generation task directly learn the distribution $p(\mathbf{y})$ over mixtures, collapsing the information needed for the separation task. In this case, we have accurate mixture modeling but no information about the individual sources. It is worth noting that approaches that model the distribution of mixtures conditioning on textual data (Schneider et al., 2023; Agostinelli et al., 2023) face the same limitations. Conversely, models for source separation (Défossez et al., 2019) either target $p(\mathbf{x}_1, \ldots, \mathbf{x}_N \mid \mathbf{y})$, conditioning on the mixture, or learn a single model $p_n(\mathbf{x}_n)$ for each source distribution (e.g., in a weakly-supervised manner) and condition on the mixture during inference (Jayaram & Thickstun, 2020; Postolache et al., 2023a). In both cases, generating mixtures is impossible. In the first case, the model inputs a mixture, which hinders the possibility of unconditional modeling, not having direct access to $p(\mathbf{x}_1, \ldots, \mathbf{x}_N)$ (or equivalently to $p(\mathbf{y})$). In the second case, while we can accurately model each source independently, all essential information about their interdependence is lost, preventing the possibility of generating coherent mixtures.

**Contribution.** Our contribution is three-fold. *(i)* First, we bridge the gap between source separation and music generation by learning $p(\mathbf{x}_1, \ldots, \mathbf{x}_N)$, the joint (prior) distribution of contextual sources (i.e., those belonging to the same song). For this purpose, we use the denoising score-matching framework to train a *Multi-Source Diffusion Model (MSDM)*. We can perform both source separation and music generation during inference by training this single model. Specifically, generation is achieved by sampling from the prior, while separation is carried out by conditioning the prior on

the mixture and then sampling from the resulting posterior distribution. *(ii)* This new formulation opens the doors to novel tasks in the generative domain, such as *source imputation*, where we create accompaniments by generating a subset of the sources given the others (e.g., play a piano track that goes well with the drums). *(iii)* Lastly, to obtain competitive results on source separation with respect to state-of-the-art regressor models (Manilow et al., 2022) on the Slakh2100 (Manilow et al., 2019) dataset, we propose a new procedure for computing the posterior score based on *Dirac delta functions*, exploiting the functional relationship between the sources and the mixture.

## 2 RELATED WORK

### 2.1 GENERATIVE MODELS FOR AUDIO

Deep generative models for audio learn, directly or implicitly, the distribution of mixtures, represented in our notation by $p(\mathbf{y})$, possibly conditioning on additional data such as text. Various general-purpose generative models, such as autoregressive models, GANs (Donahue et al., 2019), and diffusion models, have been adapted for use in the audio field.

Autoregressive models are well-established in audio modeling (van den Oord et al., 2016). Jukebox (Dhariwal et al., 2020) proposed to model musical tracks with Scalable Transformers (Vaswani et al., 2017) on hierarchical discrete representations obtained through VQ-VAEs (van den Oord et al., 2017). Furthermore, using a lyrics conditioner, this method generated tracks with vocals following the text. However, while Jukebox could model longer sequences in latent space, the audio output suffered from quantization artifacts. Newer latent autoregressive models (Borsos et al., 2023; Kreuk et al., 2023) can handle extended contexts, more coherent generations and, by incorporating residual quantization (Zeghidour et al., 2021), output more naturally sounding samples. State-of-the-art latent autoregressive models for music, such as MusicLM (Agostinelli et al., 2023), can guide generation by conditioning on textual embeddings obtained via large-scale contrastive pre-training (Manco et al., 2022; Huang et al., 2022). MusicLM can also input a melody and condition on text for style transfer. A concurrent work, SingSong (Donahue et al., 2023), introduces vocal-to-mixture accompaniment generation. Our accompaniment generation procedure differs from the latter since we perform generation at the stem level in a composable way, while the former outputs a single mixture.

DiffWave (Kong et al., 2021) and WaveGrad (Chen et al., 2021) were the first diffusion (score) based generative models in audio, tackling speech synthesis. Many subsequent models followed these preliminary works, mainly conditioned to solve particular tasks such as speech enhancement (Lu et al., 2021; Serrà et al., 2022; Sawata et al., 2023; Saito et al., 2023), audio upsampling (Lee & Han, 2021; Yu et al., 2023), MIDI-to-waveform (Mittal et al., 2021; Hawthorne et al., 2022), or spectrogram-to-MIDI generation (Cheuk et al., 2023). The first work in source-specific generation with diffusion models is CRASH (Rouard & Hadjeres, 2021). (Yang et al., 2023; Pascual et al., 2023; Liu et al., 2023) proposed text-conditioned diffusion models to generate general sounds, not focusing on restricted classes such as speech or music. Closer to our work, diffusion models targeting the musical domain are Riffusion (Forsgren & Martiros, 2022) and Moûsai (Schneider et al., 2023). Riffusion fine-tunes Stable Diffusion (Rombach et al., 2022), a large pre-trained text-conditioned vision diffusion model, over STFT magnitude spectrograms. Moûsai performs generation in a latent domain, resulting in context lengths that surpass the minute. Our score network follows the design of the U-Net proposed in Moûsai, albeit using the waveform data representation.

### 2.2 AUDIO SOURCE SEPARATION

Existing audio source separation models can be broadly classified into deterministic and generative. Deterministic source separators are parametric models that input the mixtures and systematically extract one or all sources. These models are typically trained with a regression loss (Gusó et al., 2022) on the estimated signal represented as waveform (Lluís et al., 2019; Luo & Mesgarani, 2019; Défossez et al., 2019), STFT (Takahashi et al., 2018; Choi et al., 2021), or both (Défossez, 2021). On the other hand, generative source separation models based on (independent) Bayesian inference learn a prior model for each source, thus targeting the distributions $\{p_n(\mathbf{x}_n)\}_{n=1,\ldots,N}$. The mixture is observed only during inference, where a likelihood function connects it to its constituent sources. The literature has explored different priors, such as GANs (Subakan & Smaragdis, 2018; Kong et al.,

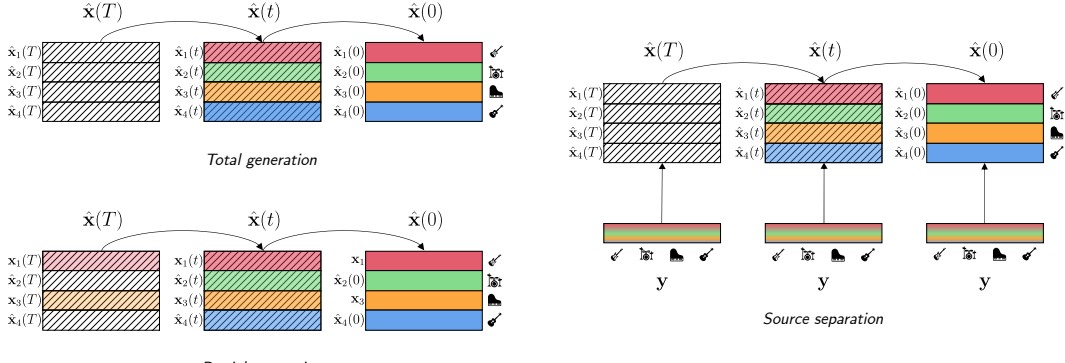

**Figure 2: Inference tasks with MSDM.** Oblique lines represent the presence of noise in the signal, decreasing from left to right, with the highest noise level at time $T$ when we start the sampling procedure. *Top-left:* We generate all stems in a mixture, obtaining a total generation. *Bottom-left:* We perform partial generation (source imputation) by fixing the sources $\mathbf{x}_1$ (Bass) and $\mathbf{x}_3$ (Piano) and generating the other two sources $\hat{\mathbf{x}}_2(0)$ (Drums) and $\hat{\mathbf{x}}_4(0)$ (Guitar). We denote with $\mathbf{x}_1(t)$ and $\mathbf{x}_3(t)$, the noisy stems obtained from $\mathbf{x}_1$ and $\mathbf{x}_3$ via the perturbation kernel in Eq. (1). *Right:* We perform source separation by conditioning the prior with a mixture $\mathbf{y}$, following Algorithm 1.

2019; Narayanaswamy et al., 2020), normalizing flows (Jayaram & Thickstun, 2020; Zhu et al., 2022), and autoregressive models (Jayaram & Thickstun, 2021; Postolache et al., 2023a).

The separation method closer to ours is NCSN-BASIS (Jayaram & Thickstun, 2020). This method was proposed for image source separation, using Langevin Dynamics to separate the mixtures with an NCSN score-based model. It employs a Gaussian likelihood function during inference, which, as we demonstrate experimentally, is sub-optimal compared to our novel Dirac-based likelihood function. The main difference between our method and other generative source separation methods (including NCSN-BASIS) is the modeling of the full joint distribution. As such, we can perform source separation and synthesize mixtures or subsets of stems with a single model.

Contextual information between sources is explicitly modeled in (Manilow et al., 2022) and (Postolache et al., 2023b). The first work models the relationship between sources by training an orderless NADE estimator, which predicts a subset of the sources while conditioning on the input mixture and the remaining sources. The subsequent study achieves universal source separation (Kavalerov et al., 2019; Wisdom et al., 2020) through adversarial training, utilizing a context-based discriminator to model the relationship between sources. Both methods are deterministic and conditioned on the mixtures architecturally. The same architectural limitation is present in diffusion-based (Scheibler et al., 2023; Lutati et al., 2023) or diffusion-inspired (Plaja-Roglans et al., 2022) conditional approaches. Our method sets itself apart as it proposes a model not constrained architecturally by a mixture conditioner, so we can also perform unconditional generation.

## 3 BACKGROUND

The foundation of our model lies in estimating the joint distribution of the sources $p(\mathbf{x}_1, \dots, \mathbf{x}_N)$. Our approach is generative because we model an unconditional distribution (the prior). The different tasks are then solved at inference time, exploiting the prior.

We employ a diffusion-based (Sohl-Dickstein et al., 2015; Ho et al., 2020) generative model trained via denoising score-matching (Song & Ermon, 2019) to learn the prior. Specifically, we present our formalism by utilizing the notation and assumptions established in (Karras et al., 2022). The central idea of score-matching (Hyvärinen, 2005; Kingma & LeCun, 2010; Vincent, 2011) is to approximate the "score" function of the target distribution $p(\mathbf{x})$, namely $\nabla_{\mathbf{x}} \log p(\mathbf{x})$, rather than the distribution itself. To effectively approximate the score in sparse data regions, denoising diffusion methods introduce controlled noise to the data and learn to remove it. Formally, the data distribution is perturbed with a Gaussian perturbation kernel:

$$p(\mathbf{x}(t) \mid \mathbf{x}(0)) = \mathcal{N}(\mathbf{x}(t); \mathbf{x}(0), \sigma^2(t)\mathbf{I}) , \tag{1}$$

where the parameter $\sigma(t)$ regulates the degree of noise added to the data. Following the authors in (Karras et al., 2022), we consider an optimal schedule given by $\sigma(t) = t$. With that choice of $\sigma(t)$, the forward evolution of a data point $\mathbf{x}(t)$ in time is described by a probability flow ODE (Song et al., 2021):

$$d\mathbf{x}(t) = -\sigma(t)\nabla_{\mathbf{x}(t)} \log p(\mathbf{x}(t)) \, dt \,. \tag{2}$$

For $t = T >> 0$, a data point $\mathbf{x}(T)$ is approximately distributed according to a Gaussian distribution $\mathcal{N}(\mathbf{x}(t); \mathbf{0}, \sigma^2(T)\mathbf{I})$, from which sampling is straightforward. Eq. (2) can be inverted in time, resulting in the following backward ODE that describes the denoising process:

$$d\mathbf{x}(t) = \sigma(t)\nabla_{\mathbf{x}(t)} \log p(\mathbf{x}(t)) \, dt \,. \tag{3}$$

Sampling can be performed integrating Eq. (3) with a standard ODE solver, starting from an initial (noisy) sample drawn from $\mathcal{N}(\mathbf{x}(t); \mathbf{0}, \sigma^2(T)\mathbf{I})$. The score function, is approximated by a neural network $S^\theta(\mathbf{x}(t), \sigma(t))$, minimizing the following score-matching loss:

$$\mathbb{E}_{t\sim\mathcal{U}([0,T])}\mathbb{E}_{\mathbf{x}(0)\sim p(\mathbf{x}(0))}\mathbb{E}_{\mathbf{x}(t)\sim p(\mathbf{x}(t)|\mathbf{x}(0))} \left\| S^\theta(\mathbf{x}(t), \sigma(t)) - \nabla_{\mathbf{x}(t)} \log p(\mathbf{x}(t) \mid \mathbf{x}(0)) \right\|_2^2 \,.$$

By expanding $p(\mathbf{x}(t) \mid \mathbf{x}(0))$ with Eq. (1), the score-matching loss simplifies to:

$$\mathbb{E}_{t\sim\mathcal{U}([0,T])}\mathbb{E}_{\mathbf{x}(0)\sim p(\mathbf{x}(0))}\mathbb{E}_{\epsilon\sim\mathcal{N}(\mathbf{0},\sigma^2(t)\mathbf{I})} \left\| D^\theta(\mathbf{x}(0) + \epsilon, \sigma(t)) - \mathbf{x}(0) \right\|_2^2 \,,$$

where we define $S^\theta(\mathbf{x}(t), \sigma(t)) =: (D^\theta(\mathbf{x}(t), \sigma(t)) - \mathbf{x}(t))/\sigma^2(t)$.

## 4 METHOD

### 4.1 MULTI-SOURCE AUDIO DIFFUSION MODELS

In our setup, we have $N$ distinct source waveforms $\{\mathbf{x}_1, \ldots, \mathbf{x}_N\}$ with $\mathbf{x}_n \in \mathbb{R}^D$ for each $n$. The sources coherently sum to a mixture $\mathbf{y} = \sum_{n=1}^{N} \mathbf{x}_n$. We also use the aggregated form $\mathbf{x} = (\mathbf{x}_1, \ldots, \mathbf{x}_N) \in \mathbb{R}^{N \times D}$. In this setting, multiple tasks can be performed: one may generate a consistent mixture $\mathbf{y}$ or separate the individual sources $\mathbf{x}$ from a given mixture $\mathbf{y}$. We refer to the first task as *generation* and the second as *source separation*. A subset of sources can also be fixed in the generation task, and the others can be generated consistently. We call this task *partial generation* or *source imputation*. Our key contribution is the ability to perform all these tasks simultaneously by training a single multi-source diffusion model (MSDM), capturing the prior $p(\mathbf{x}_1, \ldots, \mathbf{x}_N)$. The model, illustrated in Figure 1, approximates the noisy score function:

$$\nabla_{\mathbf{x}(t)} \log p(\mathbf{x}(t)) = \nabla_{(\mathbf{x}_1(t),\ldots,\mathbf{x}_N(t))} \log p(\mathbf{x}_1(t), \ldots, \mathbf{x}_N(t)) \,,$$

with a neural network:

$$S^\theta(\mathbf{x}(t), \sigma(t)) : \mathbb{R}^{N \times D} \times \mathbb{R} \to \mathbb{R}^{N \times D} \,, \tag{4}$$

where $\mathbf{x}(t) = (\mathbf{x}_1(t), \ldots, \mathbf{x}_N(t))$ denotes the sources perturbed with the Gaussian kernel in Eq. (1). We describe the three tasks (illustrated in Figure 2) using the prior distribution:

- *Total Generation.* This task requires generating a plausible mixture $\mathbf{y}$. It can be achieved by sampling the sources $\{\mathbf{x}_1, ..., \mathbf{x}_N\}$ from the prior distribution and summing them to obtain the mixture $\mathbf{y}$.

- *Partial Generation.* Given a subset of sources, this task requires generating a plausible accompaniment. We define the subset of fixed sources as $\mathbf{x}_{\mathcal{I}}$ and generate the remaining sources $\mathbf{x}_{\overline{\mathcal{I}}}$ by sampling from the conditional distribution $p(\mathbf{x}_{\overline{\mathcal{I}}} \mid \mathbf{x}_{\mathcal{I}})$.

- *Source Separation.* Given a mixture $\mathbf{y}$, this task requires isolating the individual sources that compose it. It can be achieved by sampling from the posterior distribution $p(\mathbf{x} \mid \mathbf{y})$.

### 4.2 INFERENCE

The three tasks of our method are solved during inference by discretizing the backward Eq. (3). Although different tasks require distinct score functions, they all originate directly from the prior score function in Eq. (4). We analyze each of these score functions in detail. For more details on the discretization method, refer to Section C.3.

---

**Algorithm 1** 'MSDM Dirac' sampler for source separation.

---

**Require:** $I$ number of discretization steps for the ODE, $R$ number of corrector steps, $\{\sigma_i\}_{i \in \{0,\dots,I\}}$ noise schedule, $S_{\text{churn}}$
1: Initialize $\hat{\mathbf{x}} \sim \mathcal{N}(0, \sigma_I^2 \mathbf{I})$
2: $\alpha \leftarrow \min(S_{\text{churn}}/I, \sqrt{2} - 1)$
3: **for** $i \leftarrow I$ **to** $1$ **do**
4:    **for** $r \leftarrow R$ **to** $0$ **do**
5:       $\hat{\sigma} \leftarrow \sigma_i \cdot (\alpha + 1)$
6:       $\epsilon \sim \mathcal{N}(0, \mathbf{I})$
7:       $\hat{\mathbf{x}} \leftarrow \hat{\mathbf{x}} + \sqrt{\hat{\sigma}^2 - \sigma_i^2}\epsilon$
8:       $\mathbf{z} \leftarrow [\hat{\mathbf{x}}_{1:N-1}, \mathbf{y} - \sum_{n=1}^{N-1} \hat{\mathbf{x}}_n]$
9:       **for** $n \leftarrow 1$ **to** $N - 1$ **do**
10:          $\mathbf{g}_n \leftarrow S_n^\theta(\mathbf{z}, \hat{\sigma}) - S_N^\theta(\mathbf{z}, \hat{\sigma})$
11:       **end for**
12:       $\mathbf{g} \leftarrow [\mathbf{g}_1, \dots, \mathbf{g}_{N-1}]$
13:       $\hat{\mathbf{x}}_{1:N-1} \leftarrow \hat{\mathbf{x}}_{1:N-1} + (\sigma_{i-1} - \hat{\sigma})\mathbf{g}$
14:       $\hat{\mathbf{x}} \leftarrow [\hat{\mathbf{x}}_{1:N-1}, \mathbf{y} - \sum_{n=1}^{N-1} \hat{\mathbf{x}}_n]$
15:       **if** $r > 0$ **then**
16:          $\epsilon \sim \mathcal{N}(0, \mathbf{I})$
17:          $\hat{\mathbf{x}} \leftarrow \hat{\mathbf{x}} + \sqrt{\sigma_i^2 - \sigma_{i-1}^2}\epsilon$
18:       **end if**
19:    **end for**
20: **end for**
21: **return** $\hat{\mathbf{x}}$

---

### 4.2.1 TOTAL GENERATION

The total generation task is performed by sampling from Eq. (3) using the score function in Eq. (4). The mixture is then obtained by summing over all the generated sources.

### 4.2.2 PARTIAL GENERATION

In the partial generation task, we fix a subset of source indices $\mathcal{I} \subset \{1, \dots, N\}$ and the relative sources $\mathbf{x}_\mathcal{I} := \{\mathbf{x}_n\}_{n \in \mathcal{I}}$. The goal is to generate the remaining sources $\mathbf{x}_{\overline{\mathcal{I}}} := \{\mathbf{x}_n\}_{n \in \overline{\mathcal{I}}}$ consistently, where $\overline{\mathcal{I}} = \{1, \dots, N\} - \mathcal{I}$. To do so, we estimate the gradient of the conditional distribution:

$$\nabla_{\mathbf{x}_{\overline{\mathcal{I}}}(t)} \log p(\mathbf{x}_{\overline{\mathcal{I}}}(t) \mid \mathbf{x}_\mathcal{I}(t)). \tag{5}$$

This falls into the setting of imputation or, as it is more widely known in the image domain, inpainting. We approach imputation using the method in (Song et al., 2021). The gradient in Eq. (5) is approximated as follows:

$$\nabla_{\mathbf{x}_{\overline{\mathcal{I}}}(t)} \log p([\mathbf{x}_{\overline{\mathcal{I}}}(t), \hat{\mathbf{x}}_\mathcal{I}(t)]),$$

where $\hat{\mathbf{x}}_\mathcal{I}$ is a sample from the forward process: $\hat{\mathbf{x}}_\mathcal{I}(t) \sim \mathcal{N}(\mathbf{x}_\mathcal{I}(t); \mathbf{x}_\mathcal{I}(0), \sigma(t)^2 \mathbf{I})$. The square bracket operator denotes concatenation. Approximating the score function, we write:

$$\nabla_{\mathbf{x}_{\overline{\mathcal{I}}}(t)} \log p(\mathbf{x}_{\overline{\mathcal{I}}}(t) \mid \mathbf{x}_\mathcal{I}(t)) \approx S_{\overline{\mathcal{I}}}^\theta([\mathbf{x}_{\overline{\mathcal{I}}}(t), \hat{\mathbf{x}}_\mathcal{I}(t)], \sigma(t)),$$

where $S_{\overline{\mathcal{I}}}^\theta$ denotes the entries of the score network corresponding to the sources indexed by $\overline{\mathcal{I}}$.

### 4.2.3 SOURCE SEPARATION

We view source separation as a specific instance of conditional generation, where we condition the generation process on the given mixture $\mathbf{y} = \mathbf{y}(0)$. This requires computing the score function of the posterior distribution:

$$\nabla_{\mathbf{x}(t)} \log p(\mathbf{x}(t) \mid \mathbf{y}(0)). \tag{6}$$

Standard methods for implementing conditional generation for diffusion models involve directly estimating the posterior score in Eq. (6) at training time (i.e., Classifier Free Guidance (Ho & Salimans, 2021)) or estimating the likelihood function $p(\mathbf{y}(0) \mid \mathbf{x}(t))$ and using the Bayes formula

to derive the posterior. The second approach typically involves training a separate model, often a classifier, for the likelihood score (i.e., Classifier Guidance (Dhariwal & Nichol, 2021)).

In diffusion-based generative source separation, learning a likelihood model is typically unnecessary because the relationship between $\mathbf{x}(t)$ and $\mathbf{y}(t)$ is represented by a simple function, namely the sum. A natural approach is to model the likelihood function based on such functional dependency. This is the approach taken by (Jayaram & Thickstun, 2020), where they use a Gaussian likelihood function:

$$p(\mathbf{y}(t) \mid \mathbf{x}(t)) = \mathcal{N}(\mathbf{y}(t) \mid \sum_{n=1}^{N} \mathbf{x}_n(t), \gamma^2(t)\mathbf{I}), \tag{7}$$

with the standard deviation given by a hyperparameter $\gamma(t)$. The authors argue that aligning the $\gamma(t)$ value to be proportionate to $\sigma(t)$ optimizes the outcomes of their NCSN-BASIS separator.

We present a novel approximation of the posterior score function in Eq. (6) by modeling $p(\mathbf{y}(t) \mid \mathbf{x}(t))$ as a Dirac delta function centered in $\sum_{n=1}^{N} \mathbf{x}_n(t)$:

$$p(\mathbf{y}(t) \mid \mathbf{x}(t)) = \mathbb{1}_{\mathbf{y}(t) = \sum_{n=1}^{N} \mathbf{x}_n(t)}. \tag{8}$$

The complete derivation can be found in Appendix A, and we present only the final formulation, which we call 'MSDM Dirac'. The method constrains a source, without loss of generality $\mathbf{x}_N$, by setting $\mathbf{x}_N(t) = \mathbf{y}(0) - \sum_{n=1}^{N-1} \mathbf{x}_n(t)$ and estimates:

$$\nabla_{\mathbf{x}_m(t)} \log p(\mathbf{x}(t) \mid \mathbf{y}(0)) \approx S_m^\theta((\mathbf{x}_1(t), \ldots, \mathbf{x}_{N-1}(t), \mathbf{y}(0) - \sum_{n=1}^{N-1} \mathbf{x}_n(t)), \sigma(t))$$
$$- S_N^\theta((\mathbf{x}_1(t), \ldots, \mathbf{x}_{N-1}(t), \mathbf{y}(0) - \sum_{n=1}^{N-1} \mathbf{x}_n(t)), \sigma(t)),$$

where $1 \leq m \leq N - 1$ and $S_m^\theta, S_N^\theta$ denote the entries of the score network corresponding to the $m$-th and $N$-th sources. Our approach models the limiting case wherein $\gamma(t) \to 0$ in the Gaussian likelihood function. This represents a scenario where the functional dependence between $\mathbf{x}(t)$ and $\mathbf{y}(t)$ becomes increasingly tight, thereby sharpening the conditioning on the given mixture during the generation process. The pseudo-code for the 'MSDM Dirac' source separation sampler, using the Euler ODE integrator of (Karras et al., 2022), is provided in Algorithm 1. The Euler ODE discretization logic uses the $S_{\text{churn}}$ mechanism of (Karras et al., 2022) and optional correction steps (Song et al., 2021) (see Section C.3 for more details).

The separation procedure can be additionally employed in the weakly-supervised source separation scenario, typically encountered in generative source separation (Jayaram & Thickstun, 2020; Zhu et al., 2022; Postolache et al., 2023a). This scenario pertains to cases where we know that specific audio data belongs to a particular instrument class while not having access to sets of sources sharing a context. To adapt to this scenario, we assume independence between sources $p(\mathbf{x}_1, \ldots, \mathbf{x}_N) = \prod_{n=1}^{N} p_n(\mathbf{x}_n)$ and train a separate model for each source class. We call the resulting model 'Independent Source Diffusion Model with Dirac Likelihood' ('ISDM Dirac'). We derive its formula together with formulas for the Gaussian versions 'MSDM Gaussian' and 'ISDM Gaussian' in Appendix B.

## 5 EXPERIMENTAL RESULTS

We experiment on Slakh2100 (Manilow et al., 2019), a standard dataset for music source separation. We chose Slakh2100 because it has a significantly larger quantity of data (145h) than other multi-source waveform datasets like MUSDB18-HQ (Rafii et al., 2019) (10h). The amount of data plays a decisive role in determining the quality of a generative model, making Slakh2100 a preferable choice. Nevertheless, in Appendix E we conduct a study of data efficiency on MUSDB18-HQ. Details on datasets, architecture, training, and sampling are provided in Appendix C.

### 5.1 MUSIC GENERATION

The performance of MSDM on the generative tasks is tested through subjective and objective evaluation.

**Table 1: Comparison between total generation capabilities of MSDM (Slakh2100) and an equivalent architecture trained on Slakh2100 mixtures.** Both subjective (quality and coherence, higher is better) and objective (FAD, lower is better) evaluations are shown. The quality and coherence columns refer to the average scores of the listening tests, with respective variances.

| Model | FAD $\downarrow$ | Quality $\uparrow$ | Coherence $\uparrow$ |
|---|---|---|---|
| MSDM | 6.55 | $6.51 \pm 2.19$ | $6.35 \pm 2.36$ |
| Mixture Model | 6.67 | $6.15 \pm 2.47$ | $5.67 \pm 2.60$ |

**Table 2: Quantitative and qualitative results for the partial generation task on Slakh2100.** We use both subjective (quality and density, higher is better) and objective (sub-FAD, lower is better) evaluation metrics. The sub-FAD metric is reported for all combinations of generated sources (**B**: Bass, **D**: Drums, **G**: Guitar, **P**: Piano). The quality and density columns refer to the average scores of the listening tests, with respective variances.

| Slakh2100 | B | D | G | P | BD | BG | BP | DG | DP | GP | BDG | BDP | BGP | DGP | Quality $\uparrow$ | Density $\uparrow$ |
|---|---|---|---|---|---|---|---|---|---|---|---|---|---|---|---|---|
| MSDM | 0.45 | 1.09 | 0.11 | 0.76 | 2.09 | 1.00 | 2.32 | 1.45 | 1.82 | 1.65 | 2.93 | 3.30 | 4.90 | 3.10 | $6.3 \pm 2.7$ | $6.1 \pm 2.6$ |

Subjective evaluation is done through listening tests, whose form format is reported in Appendix F. Concisely, we produce two forms, one for total generation and one for partial generation. In the first, subjects are asked to rate, from 1 to 10, the *quality* and instrument *coherence* (i.e., how the instruments sound plausible together) of 30 generated chunks, of which 15 are generated by MSDM and 15 by a model trained on mixtures (using the same diffusion architecture as MSDM). In the second one, knowing the fixed instruments, subjects are asked to rate, from 1 to 10, the *quality* and the *density* of the generated accompaniment. Namely, 'quality' tests how the full chunk sounds plausible with respect to the ground truth data, and 'density' tests how much the generated instruments are present in the chunk. We also provide examples of music and accompaniment generation[1].

For the objective evaluation of the generative tasks, we generalize the FAD protocol in Donahue et al. (2023) to our total and partial generation tasks with more than one source. Given $D_{\text{real}}$ a dataset of ground truth mixtures chunks and $\mathcal{I}$ a set indexing conditioning sources ($\emptyset$ for total generation), we build a dataset $D_{\text{gen}}$ whose elements are the sum between conditioning sources (indexed by $\mathcal{I}$) and the respective generated sources. We define the *sub-FAD* as $FAD(D_{\text{real}}, D_{\text{gen}})$. We use VGGish embeddings (Hershey et al., 2017) for computing the metric.

Results for total and partial generations are reported in Tables 1 and 2 respectively, both for subjective and objective evaluations. Results in Table 1 show a minimal difference between the model trained on mixtures and MSDM. This suggests that, given the same dataset and architecture, the generative power of MSDM is the same as the model trained on mixtures while being able to perform separation and partial generation. Table 2 shows via the subjective results that the task of partial generation can be performed with non-trivial quality. Our method being the first able to generate any combination of partial sources, does not have a competitor baseline for the objective metrics. We thus report the sub-FAD results of our method as baseline metrics for future research.

## 5.2 SOURCE SEPARATION

In order to evaluate source separation, we use the scale-invariant SDR improvement (SI-SDR$_\text{I}$) metric (Roux et al., 2019). The SI-SDR between a ground-truth source $\mathbf{x}_n$ and an estimate $\hat{\mathbf{x}}_n$ is defined as:

$$\text{SI-SDR}(\mathbf{x}_n, \hat{\mathbf{x}}_n) = 10 \log_{10} \frac{\|\alpha \mathbf{x}_n\|^2 + \epsilon}{\|\alpha \mathbf{x}_n - \hat{\mathbf{x}}_n\|^2 + \epsilon} ,$$

where $\alpha = \frac{\mathbf{x}_n^\top \hat{\mathbf{x}}_n + \epsilon}{\|\mathbf{x}_n\|^2 + \epsilon}$ and $\epsilon = 10^{-8}$. The improvement with respect to the mixture baseline is defined as $\text{SI-SDR}_\text{I} = \text{SI-SDR}(\mathbf{x}_n, \hat{\mathbf{x}}_n) - \text{SI-SDR}(\mathbf{x}_n, \mathbf{y})$.

On Slakh, we compare our supervised MSDM and weakly-supervised MSDM with the 'Demucs' (Défossez et al., 2019) and 'Demucs + Gibbs (512 steps)' regressor baselines from (Manilow et al.,

---

[1] https://gladia-research-group.github.io/multi-source-diffusion-models/

**Table 3: Quantitative results for source separation on the Slakh2100 test set.** We use the SI-SDR$_I$ as our evaluation metric (dB – higher is better). We present both the supervised ('MSDM Dirac', 'MSDM Gaussian') and weakly-supervised ('ISDM Dirac', 'ISDM Gaussian') separators and specify if a correction step is used. 'All' reports the average over the four stems.

| Model | Bass | Drums | Guitar | Piano | All |
|---|---|---|---|---|---|
| Demucs (Défossez et al., 2019; Manilow et al., 2022) | 15.77 | 19.44 | 15.30 | 13.92 | 16.11 |
| Demucs + Gibbs (512 steps) (Manilow et al., 2022) | 17.16 | 19.61 | **17.82** | **16.32** | **17.73** |
| **Dirac Likelihood** | | | | | |
| ISDM | 18.44 | 20.19 | 13.34 | 13.25 | 16.30 |
| ISDM (correction) | **19.36** | **20.90** | 14.70 | 14.13 | 17.27 |
| MSDM | 16.21 | 17.47 | 12.71 | 13.29 | 14.92 |
| MSDM (correction) | 17.12 | 18.68 | 15.38 | 14.73 | 16.48 |
| **Gaussian Likelihood** (Jayaram & Thickstun, 2020) | | | | | |
| ISDM | 13.48 | 18.09 | 11.93 | 11.17 | 13.67 |
| ISDM (correction) | 14.27 | 19.10 | 12.74 | 12.20 | 14.58 |
| MSDM | 12.53 | 16.82 | 12.98 | 9.29 | 12.90 |
| MSDM (correction) | 13.93 | 17.92 | 14.19 | 12.11 | 14.54 |

2022), the state-of-the-art for supervised music source separation on Slakh2100, aligning with the evaluation procedure of (Manilow et al., 2022). We evaluate over the test set of Slakh2100, using chunks of 4 seconds in length (with an overlap of two seconds) and filtering out silent chunks and chunks consisting of only one source, given the poor performance of SI-SDR$_I$ on such segments. We report results comparing our Dirac score posterior with the Gaussian score posterior of (Jayaram & Thickstun, 2020), using the best parameters of the ablations in Appendix D and 150 inference steps.

Results are reported in Table 3 and show that: *(i)* The Dirac likelihood improves overall results, even outperforming the state of the art when applied to ISDM on Bass and Drums *(ii)* adding a correction step is beneficial *(iii)* MSDM with Dirac likelihood and one step of correction gives results comparable with the state of the art and superior to standard Demucs overall. We stress again that, while the baselines can perform the separation task alone, MSDM can also perform generative tasks.

## 6 CONCLUSIONS

We have presented a general method, based on denoising score-matching, for source separation, mixture generation, and accompaniment generation in the musical domain. Our approach utilizes a single neural network trained once, with tasks differentiated during inference. Moreover, we have defined a new sampling method for source separation. We quantitatively tested the model on source separation, obtaining results comparable to state-of-the-art regressor models. We qualitatively and quantitatively tested the model on total and partial generation. Our model's ability to handle both total and partial generation and source separation positions it as a significant step toward the development of general audio models. This flexibility paves the way for more advanced music composition tools, where users can easily control and manipulate individual sources within a mixture.

### 6.1 LIMITATIONS AND FUTURE WORK

The amount of available contextual data constrains the performance of our model. To address this, pre-separating mixtures and training on the separations, as demonstrated in (Donahue et al., 2023), may prove beneficial. Additionally, it would be intriguing to explore the possibility of extending our method to situations where the sub-signals are not related by addition but rather by a known but different function. Finally, future work could adapt the model to jointly model MIDI information (for example, extracted from sources (Lin et al., 2021)) for further control.

## ACKNOWLEDGEMENTS

The authors were partially supported by the ERC grant no. 802554 (SPECGEO), PRIN 2020 project no. 2020TA3K9N (LEGO.AI), PRIN 2022 project no. 2022AL45R2 (EYE-FI.AI, CUP H53D2300350-0001), and PNRR MUR project no. PE0000013-FAIR.

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

## A DERIVATION OF MSDM DIRAC POSTERIOR SCORE FOR SOURCE SEPARATION

We prove the main result of Section 4.2.3. We condition the generative model over the mixture $\mathbf{y}(0) = \mathbf{y}$. As such, we compute the posterior:

$$p(\mathbf{x}(t) \mid \mathbf{y}(0)) = \int_{\mathbf{y}(t)} p(\mathbf{x}(t), \mathbf{y}(t) \mid \mathbf{y}(0))\mathrm{d}\mathbf{y}(t) = \int_{\mathbf{y}(t)} p(\mathbf{x}(t) \mid \mathbf{y}(t), \mathbf{y}(0))p(\mathbf{y}(t) \mid \mathbf{y}(0))\mathrm{d}\mathbf{y}(t).$$

The first equality is given by marginalizing over $\mathbf{y}(t)$ and the second by the chain rule. Following Eq. (50) in Song et al. (2021), we can eliminate the dependency on $\mathbf{y}(0)$ from the first term, obtaining the approximation:

$$p(\mathbf{x}(t) \mid \mathbf{y}(0)) \approx \int_{\mathbf{y}(t)} p(\mathbf{x}(t) \mid \mathbf{y}(t))p(\mathbf{y}(t) \mid \mathbf{y}(0))\mathrm{d}\mathbf{y}(t). \tag{9}$$

We compute $p(\mathbf{y}(t) \mid \mathbf{y}(0))$, using the chain rule after marginalizing over $\mathbf{x}(0)$ and $\mathbf{x}(t)$:

$$p(\mathbf{y}(t) \mid \mathbf{y}(0)) = \int_{\mathbf{x}(0), \mathbf{x}(t)} p(\mathbf{y}(t), \mathbf{x}(t), \mathbf{x}(0) \mid \mathbf{y}(0))\mathrm{d}\mathbf{x}(0)\mathrm{d}\mathbf{x}(t)$$

$$= \int_{\mathbf{x}(0), \mathbf{x}(t)} p(\mathbf{y}(t) \mid \mathbf{x}(t), \mathbf{x}(0), \mathbf{y}(0))p(\mathbf{x}(t) \mid \mathbf{x}(0), \mathbf{y}(0))p(\mathbf{x}(0) \mid \mathbf{y}(0))\mathrm{d}\mathbf{x}(0)\mathrm{d}\mathbf{x}(t).$$

By the Markov property of the forward diffusion process, $\mathbf{y}(t)$ is conditionally independent from $\mathbf{x}(0)$ given $\mathbf{x}(t)$ and we drop again the conditioning on $\mathbf{y}(0)$ from the first two terms, following Eq. (50) in Song et al. (2021). As such, we have:

$$p(\mathbf{y}(t) \mid \mathbf{y}(0)) \approx \int_{\mathbf{x}(0), \mathbf{x}(t)} p(\mathbf{x}(0) \mid \mathbf{y}(0))p(\mathbf{x}(t) \mid \mathbf{x}(0))p(\mathbf{y}(t) \mid \mathbf{x}(t))\mathrm{d}\mathbf{x}(0)\mathrm{d}\mathbf{x}(t). \tag{10}$$

We model the likelihood function $p(\mathbf{y}(t) \mid \mathbf{x}(t))$ with the Dirac delta function in Eq. (8). The posterior $p(\mathbf{x}(0) \mid \mathbf{y}(0))$ is obtained via Bayes theorem substituting the likelihood:

$$p(\mathbf{x}(0) \mid \mathbf{y}(0)) = \frac{p(\mathbf{x}(0))\mathbb{1}_{\mathbf{y}(0)=\sum_{n=1}^{N}\mathbf{x}_n(0)}}{p(\mathbf{y}(0))} = \begin{cases} \frac{p(\mathbf{x}(0))}{p(\mathbf{y}(0))} & \text{if } \sum_{n=1}^{N}\mathbf{x}_n(0) = \mathbf{y}(0) \\ 0 & \text{otherwise} \end{cases}$$

We substitute it in Eq. (10), together with Eq. (1) and Eq. (8), obtaining:

$$\int_{\mathbf{x}(0):\sum_{n=1}^{N}\mathbf{x}(0)=\mathbf{y}(0)} \frac{p(\mathbf{x}(0))}{p(\mathbf{y}(0))} \int_{\mathbf{x}(t)} \mathcal{N}(\mathbf{x}(t); \mathbf{x}(0), \sigma^2(t)\mathbf{I})\mathbb{1}_{\mathbf{y}(t)=\sum_{n=1}^{N}\mathbf{x}_n(t)}\mathrm{d}\mathbf{x}(t)\mathrm{d}\mathbf{x}(0). \tag{11}$$

We sum over the first $N-1$ sources in the inner integral, setting $\mathbf{x}_N(t) = \mathbf{y}(t) - \sum_{n=1}^{N-1}\mathbf{x}_n(t)$:

$$\int_{\mathbf{x}_{1:N-1}(t)} \mathcal{N}\left(\mathbf{x}_{1:N-1}(t), \mathbf{y}(t) - \sum_{n=1}^{N-1}\mathbf{x}_n(t); \mathbf{x}(0), \sigma^2(t)\mathbf{I}\right)\mathrm{d}\mathbf{x}_{1:N-1}(t) \tag{12}$$

$$= \int_{\mathbf{x}_{1:N-1}(t)} \prod_{n=1}^{N-1} \mathcal{N}(\mathbf{x}_n(t); \mathbf{x}_n(0), \sigma^2(t)\mathbf{I})\mathcal{N}\left(\mathbf{y}(t) - \sum_{n=1}^{N-1}\mathbf{x}_n(t); \mathbf{x}_N(0), \sigma^2(t)\mathbf{I}\right)\mathrm{d}\mathbf{x}_{1:N-1}(t)$$

$$= \mathcal{N}\left(\mathbf{y}(t); \sum_{n=1}^{N}\mathbf{x}_n(0), N\sigma^2(t)\mathbf{I}\right). \tag{13}$$

The second equality is obtained by factorizing the Gaussian, which has diagonal covariance matrix, while the last equality is obtained by iterative application of the convolution theorem Katznelson (2004). We substitute Eq. (13) in Eq. (11), obtaining:

$$
\begin{aligned}
p(\mathbf{y}(t) \mid \mathbf{y}(0)) &\approx \int_{\mathbf{x}(0):\sum_{n=1}^{N} \mathbf{x}_n(0)=\mathbf{y}(0)} \frac{p(\mathbf{x}(0))}{p(\mathbf{y}(0))} \mathcal{N}(\mathbf{y}(t); \sum_{n=1}^{N} \mathbf{x}_n(0), N\sigma^2(t)\mathbf{I}) \mathrm{d}\mathbf{x}(0) \\
&= \mathcal{N}(\mathbf{y}(t); \mathbf{y}(0), N\sigma^2(t)\mathbf{I}) \int_{\mathbf{x}(0):\sum_{n=1}^{N} \mathbf{x}_n(0)=\mathbf{y}(0)} \frac{p(\mathbf{x}(0))}{p(\mathbf{y}(0))} \mathrm{d}\mathbf{x}(0) \\
&= \mathcal{N}(\mathbf{y}(t); \mathbf{y}(0), N\sigma^2(t)\mathbf{I}) \,.
\end{aligned}
\tag{14}
$$

At this point, we apply Bayes theorem in Eq. (9), substituting the Dirac likelihood:

$$
p(\mathbf{x}(t) \mid \mathbf{y}(0)) \approx \int_{\mathbf{y}(t)} \frac{p(\mathbf{x}(t))p(\mathbf{y}(t) \mid \mathbf{x}(t))}{p(\mathbf{y}(t))} p(\mathbf{y}(t) \mid \mathbf{y}(0)) \mathrm{d}\mathbf{y}(t)
\tag{15}
$$

$$
= \int_{\mathbf{y}(t)} \frac{p(\mathbf{x}(t))\mathbb{1}_{\mathbf{y}(t)=\sum_{n=1}^{N} \mathbf{x}_n(t)}}{p(\mathbf{y}(t))} p(\mathbf{y}(t) \mid \mathbf{y}(0)) \mathrm{d}\mathbf{y}(t)
\tag{16}
$$

$$
= \frac{p(\mathbf{x}(t))}{p(\sum_{n=1}^{N} \mathbf{x}_n(t))} p(\sum_{n=1}^{N} \mathbf{x}_n(t) \mid \mathbf{y}(0)) \,.
\tag{17}
$$

Estimating Eq. (17), however, requires knowledge of the mixture density $p(\sum_{n=1}^{N} \mathbf{x}_n(t))$, which we do not acknowledge. As such, we approximate Eq. (16) with Monte Carlo, using the mean of $p(\mathbf{y}(t) \mid \mathbf{y}(0))$, namely $\mathbf{y}(0)$ (see Eq. (14)), obtaining:

$$
p(\mathbf{x}(t) \mid \mathbf{y}(0)) \approx \frac{p(\mathbf{x}(t))\mathbb{1}_{\mathbf{y}(0)=\sum_{n=1}^{N} \mathbf{x}_n(t)}}{p(\mathbf{y}(0))} = \begin{cases} \frac{p(\mathbf{x}(t))}{p(\mathbf{y}(0))} & \text{if } \sum_{n=1}^{N} \mathbf{x}_n(t) = \mathbf{y}(0) \\ 0 & \text{otherwise} \end{cases}
\tag{18}
$$

Similar to how we constrained the integral in Eq. (12), we parameterize the posterior, without loss of generality, using the first $N-1$ sources $\tilde{\mathbf{x}}(t) = (\mathbf{x}_1(t), \ldots, \mathbf{x}_{N-1}(t))$. The last source is constrained setting $\mathbf{x}_N(t) = \mathbf{y}(0) - \sum_{n=1}^{N-1} \mathbf{x}_n(t)$ and the parameterization is defined as:

$$
F(\tilde{\mathbf{x}}(t)) = F(\mathbf{x}_1(t), \ldots, \mathbf{x}_{N-1}(t)) = (\mathbf{x}_1(t), \ldots, \mathbf{x}_{N-1}(t), \mathbf{y}(0) - \sum_{n=1}^{N-1} \mathbf{x}_n(t)) \,.
\tag{19}
$$

Plugging Eq. (19) in Eq. (18) we obtain the parameterized posterior:

$$
p(F(\tilde{\mathbf{x}}(t)) \mid \mathbf{y}(0)) \approx \frac{p(F(\tilde{\mathbf{x}}(t)))}{p(\mathbf{y}(0))}
\tag{20}
$$

At this point, we compute the gradient of the logarithm of Eq. (20) with respect to $\tilde{\mathbf{x}}(t)$:

$$
\begin{aligned}
\nabla_{\tilde{\mathbf{x}}(t)} \log p(F(\tilde{\mathbf{x}}(t)) \mid \mathbf{y}(0)) &\approx \nabla_{\tilde{\mathbf{x}}(t)} \log \frac{p(F(\tilde{\mathbf{x}}(t)))}{p(\mathbf{y}(0))} \\
&= \nabla_{\tilde{\mathbf{x}}(t)} \log p(F(\tilde{\mathbf{x}}(t))) - \nabla_{\tilde{\mathbf{x}}(t)} \log p(\mathbf{y}(0)) \\
&= \nabla_{\tilde{\mathbf{x}}(t)} \log p(F(\tilde{\mathbf{x}}(t))) \,.
\end{aligned}
\tag{21}
$$

Using the chain-rule for differentiation on Eq. (21) we have:

$$
\nabla_{\tilde{\mathbf{x}}(t)} \log p(F(\tilde{\mathbf{x}}(t)) \mid \mathbf{y}(0)) \approx \nabla_{F(\tilde{\mathbf{x}}(t))} \log p(F(\tilde{\mathbf{x}}(t))) J_F(\tilde{\mathbf{x}}(t)),
\tag{22}
$$

where $J_F(\tilde{\mathbf{x}}(t)) \in \mathbb{R}^{(N \times D) \times ((N-1) \times D)}$ is the Jacobian of $F$ computed in $\tilde{\mathbf{x}}(t)$, equal to:

$$
J_F(\tilde{\mathbf{x}}(t)) = \begin{bmatrix} \mathbf{I} & \mathbf{0} & \ldots & \mathbf{0} \\ \mathbf{0} & \mathbf{I} & \ldots & \mathbf{0} \\ \vdots & \vdots & \ddots & \vdots \\ \mathbf{0} & \mathbf{0} & \ldots & \mathbf{I} \\ -\mathbf{I} & -\mathbf{I} & \ldots & -\mathbf{I} \end{bmatrix}
$$

The gradient with respect to a source $\mathbf{x}_m(t)$ with $1 \le m \le N - 1$ in Eq. (22) is thus equal to:

$$\nabla_{\mathbf{x}_m(t)} \log p(F(\tilde{\mathbf{x}}(t)) \mid \mathbf{y}(0)) \approx [\nabla_{F(\tilde{\mathbf{x}}(t))} \log p(F(\tilde{\mathbf{x}}(t)))]_m$$
$$- [\nabla_{F(\tilde{\mathbf{x}}(t))} \log p(F(\tilde{\mathbf{x}}(t)))]_N \,,$$

where we index the components of the $m$-th and $N$-th sources in $\nabla_{F(\tilde{\mathbf{x}}(t))} \log p(F(\tilde{\mathbf{x}}(t)))$. Finally, we replace the gradients with the score networks:

$$\nabla_{\mathbf{x}_m(t)} \log p(F(\tilde{\mathbf{x}}(t))|\mathbf{y}(0)) \approx S_m^\theta((\mathbf{x}_1(t), \dots, \mathbf{x}_{N-1}(t), \mathbf{y}(0) - \sum_{n=1}^{N-1} \mathbf{x}_n(t)), \sigma(t))$$

$$- S_N^\theta((\mathbf{x}_1(t), \dots, \mathbf{x}_{N-1}(t), \mathbf{y}(0) - \sum_{n=1}^{N-1} \mathbf{x}_n(t)), \sigma(t)) \,, \qquad (23)$$

where $S_m^\theta$ and $S_N^\theta$ are the entries of the score network corresponding to the $m$-th and $N$-th sources.

# B DERIVATION OF GAUSSIAN AND WEAKLY-SUPERVISED POSTERIOR SCORES FOR SOURCE SEPARATION

In this Section we derive the formulas for 'MSDM Gaussian', 'ISDM Dirac' and 'ISDM Gaussian'. We first adapt the Gaussian posterior introduced in Jayaram & Thickstun (2020) to continuous-time score-based diffusion models Karras et al. (2022). We plug the Gaussian likelihood function (Eq. (7)) into Eq. (15), obtaining:

$$p(\mathbf{x}(t) \mid \mathbf{y}(0)) \approx \int_{\mathbf{y}(t)} \frac{p(\mathbf{x}(t)) \mathcal{N}(\mathbf{y}(t); \sum_{n=1}^N \mathbf{x}_n(t), \gamma^2(t)\mathbf{I})}{p(\mathbf{y}(t))} p(\mathbf{y}(t) \mid \mathbf{y}(0)) \mathrm{d}\mathbf{y}(t) \qquad (24)$$

Following Jayaram & Thickstun (2020), $\mathbf{y}(t)$ is not re-sampled during inference and is always set to $\mathbf{y}(0)$. As such, we perform Monte Carlo in Eq. (24) with $\mathbf{y}(0)$, the mean of $p(\mathbf{y}(t) \mid \mathbf{y}(0))$ (see Eq. (14)), obtaining:

$$p(\mathbf{x}(t) \mid \mathbf{y}(0)) \approx \frac{p(\mathbf{x}(t)) \mathcal{N}(\mathbf{y}(0); \sum_{n=1}^N \mathbf{x}_n(t), \gamma^2(t)\mathbf{I})}{p(\mathbf{y}(0))} \,. \qquad (25)$$

At this point, we compute the gradient of the logarithm of Eq. (25) with respect to $\mathbf{x}_m(t)$:

$$\nabla_{\mathbf{x}_m(t)} \log p(\mathbf{x}(t) \mid \mathbf{y}(0)) \approx \nabla_{\mathbf{x}_m(t)} \log \frac{p(\mathbf{x}(t)) \mathcal{N}(\mathbf{y}(0); \sum_{n=1}^N \mathbf{x}_n(t), \gamma^2(t)\mathbf{I})}{p(\mathbf{y}(0))}$$

$$= \nabla_{\mathbf{x}_m(t)} \log p(\mathbf{x}(t)) + \nabla_{\mathbf{x}_m(t)} \log \mathcal{N}(\mathbf{y}(0); \sum_{n=1}^N \mathbf{x}_n(t), \gamma^2(t)\mathbf{I})$$

$$= \nabla_{\mathbf{x}_m(t)} \log p(\mathbf{x}(t)) - \frac{1}{2\gamma^2(t)} \nabla_{\mathbf{x}_m(t)} \|\mathbf{y}(0) - \sum_{n=1}^N \mathbf{x}_n(t)\|_2^2$$

$$= \nabla_{\mathbf{x}_m(t)} \log p(\mathbf{x}(t)) - \frac{1}{\gamma^2(t)} (\mathbf{y}(0) - \sum_{n=1}^N \mathbf{x}_n(t)) \,. \qquad (26)$$

We obtain the 'MSDM Gaussian' posterior score by replacing the contextual prior with the score network:

$$\nabla_{\mathbf{x}_m(t)} \log p(\mathbf{x}(t) \mid \mathbf{y}(0)) \approx S_m^\theta((\mathbf{x}_1(t), \dots, \mathbf{x}_N(t)), \sigma(t)) - \frac{1}{\gamma^2(t)} (\mathbf{y}(0) - \sum_{n=1}^N \mathbf{x}_n(t)) \,. \qquad (27)$$

The weakly-supervised posterior scores are obtained by approximating:

$$p(\mathbf{x}_1(t), \dots, \mathbf{x}_N(t)) \approx \prod_{n=1}^N p_n(\mathbf{x}_n(t)) \,,$$

| Model | Inference Time (s) | # of parameters |
|---|---|---|
| Demucs | $0.111_{\pm 0.071}$ | 40M |
| Demucs + Gibbs (512 steps) | $0.111_{\pm 0.071} \times 512 = 56.832_{\pm 36.352}$ | $\sim 40$M |
| Demucs + Gibbs (256 steps) | $0.111_{\pm 0.071} \times 256 = 28.416_{\pm 18.176}$ | $\sim 40$M |
| ISDM (correction) | $4.6_{\pm 0.345} \times 4 = 18.4_{\pm 1.38}$ | 405M $\times 4$ |
| MSDM (correction) | $4.6_{\pm 0.345}$ | 405M |

**Table 4: Inference times for a single 12s long separation and number of parameters for each model in Table 3.** We report Demucs + Gibbs (256 steps) since the minimum number of steps that makes the SI-SDR$_I$ over all instruments (17.59 dB) greater than ISDM. While ISDM and MSDM are not time-competitive to Demucs, they are more time-efficient than Demucs + Gibbs (256 and 512 steps).

where $p_n$ are estimated with independent score functions $S_n^\theta$. In the contextual samplers in Eq. (23) ('MSDM Dirac') and Eq. (27) ('MSDM Gaussian'), $S_n^\theta((\mathbf{x}_1(t), \ldots, \mathbf{x}_N(t)), \sigma(t))$ refers to a slice of the full score network on the components of the $n-$th source. In the weakly-supervised cases, $S_n^\theta$ is an individual function. To obtain the 'ISDM Dirac' posterior score, we factorize the prior in Eq. (21), then use the chain rule of differentiation, as in Appendix A, to obtain:

$$\nabla_{\mathbf{x}_m(t)} \log p(F(\tilde{\mathbf{x}}(t)) \mid \mathbf{y}(0)) \approx \nabla_{\mathbf{x}_m(t)} \log p_m(\mathbf{x}_m(t)) + \nabla_{\mathbf{x}_m(t)} \log p_N(\mathbf{y}(0) - \sum_{n=1}^{N-1} \mathbf{x}_n(t))$$

$$\approx S_m^\theta(\mathbf{x}_m(t), \sigma(t)) - S_N^\theta(\mathbf{y}(0) - \sum_{n=1}^{N-1} \mathbf{x}_n(t), \sigma(t)).$$

We obtain the 'ISDM Gaussian' posterior score by factorizing the joint prior in Eq. (26):

$$\nabla_{\mathbf{x}_m(t)} \log p(\mathbf{x}(t) \mid \mathbf{y}(0)) \approx S_m^\theta(\mathbf{x}_m(t), \sigma(t)) - \frac{1}{\gamma^2(t)}(\mathbf{y}(0) - \sum_{n=1}^{N} \mathbf{x}_n(t)).$$

## C  EXPERIMENTAL SETUP

### C.1  DATASET

We perform experiments mainly on Slakh2100 (Manilow et al., 2019), a standard dataset for music source separation. Slakh2100 is a collection of multi-track waveform music data synthesized from MIDI files using virtual instruments of professional quality. The dataset comprises 2100 tracks, with a distribution of 1500 tracks for training, 375 for validation, and 225 for testing. Each track is accompanied by its stems, which belong to 31 instrumental classes. For a fair comparison, we only used the four most abundant classes as in (Manilow et al., 2022), namely Bass, Drums, Guitar, and Piano; these instruments are present in the majority of the songs: 94.7% (Bass), 99.3% (Drums), 100.0% (Guitar), and 99.3% (Piano).

In Appendix E, we experiment on MUSDB18-HQ Rafii et al. (2019), a benchmark dataset for the music source separation task. It contains 150 tracks, with 100 allocated for training and 50 for testing, amounting to roughly 10 hours of professional-grade audio. Each piece within the dataset is separated into the stems: Bass, Drums, Vocals, and Other, with the latter encompassing any elements not included in the categories above.

### C.2  ARCHITECTURE AND TRAINING

The implementation of the score network is based on a time domain (non-latent) unconditional version of Moûsai (Schneider et al., 2023).

We used the publicly available repository `audio-diffusion-pytorch/v0.0.43`[2]. The score network is a U-Net Ronneberger et al. (2015) comprised of encoder, bottleneck, and de-

---

[2]https://github.com/archinetai/audio-diffusion-pytorch/tree/v0.0.43

**Table 5: Hyperparameter search for source separation.** We use 'MSDM Dirac' (top-left), 'ISDM Dirac' (bottom-left), 'MSDM Gaussian' (top-right) and 'ISDM Gaussian' (bottom-right) posteriors. We report the SI-SDR$_I$ values in dB (higher is better) averaged over all instruments (Bass, Drums, Piano, Guitar).

| | $S_{\text{churn}}$ | Dirac Likelihood | | | | Gaussian Likelihood | | | | | | |
|---|---|---|---|---|---|---|---|---|---|---|---|---|
| | | Constrained Source | | | | $\gamma(t)$ | | | | | | |
| | | Bass | Drums | Guitar | Piano | $0.25\sigma(t)$ | $0.5\sigma(t)$ | $0.75\sigma(t)$ | $1\sigma(t)$ | $1.25\sigma(t)$ | $1.5\sigma(t)$ | $2\sigma(t)$ |
| MSDM | 0 | 4.41 | 5.05 | 3.28 | 2.87 | -41.54 | 6.37 | 6.05 | 5.67 | 5.729 | 5.13 | 4.33 |
| | 1 | 7.90 | 8.18 | 7.03 | 7.05 | -47.24 | 6.79 | 6.51 | 6.15 | 6.19 | 5.66 | 4.45 |
| | 20 | **14.29** | 12.99 | 12.19 | 11.69 | -47.17 | 11.07 | 10.51 | 9.43 | 10.19 | 9.18 | 7.58 |
| | 40 | 14.28 | 13.02 | 5.51 | 4.78 | -47.17 | -36.92 | **12.48** | 11.25 | 11.87 | 10.80 | 9.03 |
| ISDM | 0 | 5.05 | 3.69 | -2.50 | 6.93 | -45.46 | 7.12 | 6.50 | 5.78 | 5.02 | 4.49 | 3.69 |
| | 1 | 9.23 | 8.57 | 7.28 | 9.20 | -47.54 | 7.57 | 7.20 | 6.32 | 5.35 | 4.82 | 3.83 |
| | 20 | 15.35 | 15.08 | 13.20 | 15.36 | -46.86 | 12.89 | 12.21 | 10.87 | 9.32 | 8.32 | 6.47 |
| | 40 | **17.26** | 15.77 | 15.30 | 14.98 | -46.86 | -35.97 | **14.09** | 12.82 | 10.85 | 10.02 | 8.26 |
| | 60 | 16.21 | 15.57 | 15.51 | 14.20 | -46.80 | -46.85 | 14.06 | 12.57 | 11.83 | 10.81 | 9.24 |

coder with skip connections between the encoder and the decoder. The encoder has six layers comprising two convolutional ResNet blocks, followed by multi-head attention in the final three layers. The signal sequence is downsampled in each layer by a factor of 4. The number of channels in the encoder layers is [256, 512, 1024, 1024, 1024, 1024]. The bottleneck consists of a ResNet block, followed by self-attention, and another ResNet block (all with 1024 channel layers). The decoder follows a reverse symmetric structure with respect to the encoder. We employ `audio-diffusion-pytorch-trainer`[3] for training. We downsample data to 22kHz and train the score network with four stacked mono channels for MSDM (i.e., one for each stem) and one mono channel for each model in ISDM, using a context length of $\sim 12$ seconds. All our models were trained until convergence on an NVIDIA RTX A6000 GPU with 24 GB of VRAM. We trained all our models using Adam Kingma & Ba (2015), with a learning rate of $10^{-4}$, $\beta_1 = 0.9$, $\beta_2 = 0.99$ and a batch size of 16.

We report inference times and number of parameters of the various models in Table 4.

### C.3 THE SAMPLER

We use a first-order ODE integrator based on the Euler method and introduce stochasticity following (Karras et al., 2022). The amount of stochasticity is controlled by the parameter $S_{\text{churn}}$. As shown in Appendix D and explained in detail in (Karras et al., 2022), stochasticity significantly improves sample quality. We implemented a correction mechanism (Song et al., 2021; Jayaram & Thickstun, 2020) iterating for $R$ steps after each prediction step $i$, adding additional noise and re-optimizing with the score network fixed at $\sigma_i$. As per Karras et al. (2022), we adopt a non-linear schedule for time discretization that gives more importance to lower noise levels. It is defined as:

$$t_i = \sigma_i = \sigma_{\text{max}}^{\frac{1}{\rho}} + \frac{i}{I-1}(\sigma_{\text{min}}^{\frac{1}{\rho}} - \sigma_{\text{max}}^{\frac{1}{\rho}})^{\rho} \, ,$$

where $0 \leq i < I$, with $I$ the number of discretization steps. We set $\sigma_{\text{min}} = 10^{-4}$, $\sigma_{\text{max}} = 1$, $\rho = 7$.

## D HYPERPARAMETER SEARCH FOR SOURCE SEPARATION

We conduct a hyperparameter search over $S_{\text{churn}}$ to evaluate the importance of stochasticity in source separation over a fixed subset of 100 chunks of the Slakh2100 test set, each spanning 12 seconds (selected randomly). To provide a fair comparison between the Dirac ('MSDM Dirac', 'ISDM Dirac') and Gaussian ('MSDM Gaussian', 'ISDM Gaussian') posterior scores, we execute a search over their specific hyperparameters, namely the constrained source for the Dirac separators and the

---

[3]`https://github.com/archinetai/audio-diffusion-pytorch-trainer/tree/` `79229912`

**Table 6: Comparison of results of MSDM and Demucs v2 (Défossez et al., 2019)**. We report the SI-SDR$_I$ values in dB (higher is better). The network is the same as the one trained on Slakh2100, except that the sampling rate is 44kHz and is trained on 6s long chunks.

| Model | Tested on MUSDB | | | Finetuned on MUSDB | | | Trained on MUSDB | | | | |
|---|---|---|---|---|---|---|---|---|---|---|---|
| | Bass | Drums | All | Bass | Drums | All | Bass | Drums | Other | Vocals | All |
| Demucs v2 | - | - | - | - | - | - | 13.28 | 11.53 | 8.59 | 16.80 | 12.55 |
| MSDM | -0.83 | -0.94 | -0.88 | 3.46 | 5.03 | 4.25 | 4.87 | 3.28 | 1.97 | 6.83 | 4.24 |

$\gamma(t)$ coefficient for the Gaussian separators. Results are illustrated in Table 5. We observe that: *(i)* stochasticity proves beneficial for all separators, given that the highest values of SI-SDR$_I$ are achieved with $S_{\text{churn}} = 20$ and $S_{\text{churn}} = 40$, *(ii)* using the Dirac likelihood we obtain higher values of SI-SDR$_I$ with respect to the Gaussian likelihood, both with the MSDM and ISDM separators, and *(iii)* the ISDM separators perform better than the contextual MSDM separators (at the expense of not being able to perform total and partial generation).

## E  DATA EFFICIENCY STUDY: RESULTS ON MUSDB

We report in Table 6 the results of MSDM and Demucs v2 (Défossez et al., 2019) on the MUSDB18-HQ test set Rafii et al. (2019). We try three different strategies, we first check the out-of-distribution ability of the model trained on Slakh2100 by testing directly on MUSDB18-HQ. Then, we try finetuning the model trained on Slakh2100 on MUSDB18-HQ, and finally, we train directly on MUSDB18-HQ. Since the only stems shared between MUSDB18-HQ and Slakh2100 are 'Bass' and 'Drums', the first and second strategies can be tested only on these two stems.

## F  SUBJECTIVE EVALUATION

The details of the listening test are explained in Figure 3.

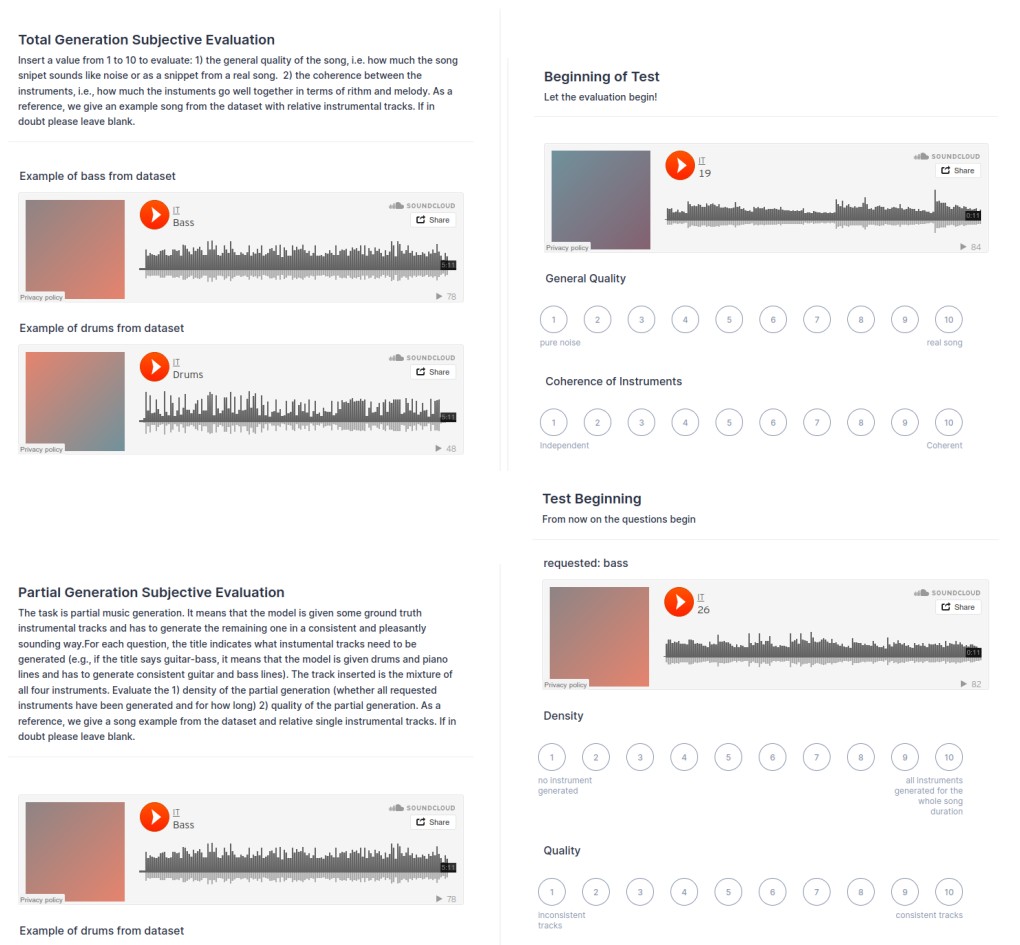

**Figure 3: Snippets from the subjective evaluation form.** The first row is relative to total generation, where people were asked to evaluate 30 songs, 15 of which were generated by the mixture model and 15 by MSDM. Thirty-two people answered the survey. The second row is relative to partial generation. Subjects were asked to evaluate 15 songs. A random subset of sources is fixed for each song, and MSDM generates the other. The requested sources are explicitly stated above the song (e.g., in the snippet, the model has generated only the Bass stem). Twenty-one subjects answered.

