# OpenReview forum: "Multi-Source Diffusion Models for Simultaneous Music Generation and Separation"
_ICLR.cc/2024/Conference — ICLR 2024 oral_

### Official Review · Reviewer_yjJy · 2023-10-23

**Soundness:** 3 good
**Presentation:** 4 excellent
**Contribution:** 3 good
**Rating:** 8
**Confidence:** 3

**Summary:**

This paper proposes a unified approach (called Multi-Source Diffusion Model) to solve audio generation, source inputation and source separation. In training time, the diffusion model learns the joint distribution of the audio mixture and solves the three tasks using different inference methods. Audio generation is done via directly sampling the prior. Source inputation is done using the inpainting technique in diffusion models. The source separation is a tailored method: reconstructing audio signals under the constraint that audio mixture is the summation of each audio tracks. Experiment results show the model is successful in all three tasks.

**Strengths:**

1. Originality. The idea of using a generative way to view music source separation is so natural and experiments show diffusion model can perform pretty well. The diffusion sampling process under summation constraint is borrowed from works in CV but it is a natural fit for source separation tasks.
2. Quality: theory is presented clearly and experiments are effective.
3. Clarity: paper is well-written. High-level idea and detail are both clear.
4. Significance: the result is convincing according to the demo in the supplementary material.

**Weaknesses:**

The paper is well-written. I have some minor comments. If it can be clarified, the significance of the work will be increased.

1. The idea of using a generative way to model audio and solve multiple audio tasks in one model is not a new idea. For example, we have a unified VAE approach to do source separation and transcription [1]. Of course the current work is novel but it would be better to include those studies and compare the approaches in related work.
2. What is the novelty compared to NCSN-BASIS? Any intuition for MSDM Dirac? Is it a marginal improvement from existing methods, or not?
3. The introduction of ISDM method is not clear. Is it a baseline method where the assumptions are obviously wrong? Is it another valid approach? Compared to MSDM, is there a trade-off in terms of generation quality and separation performance?
4. The introduction of correction step is okay. Since it is evaluated in the experiment section, could you explain more in the paper? What is the statement to be expected prior to the experiment regarding correction step?

[1] Liwei Lin, Gus Xia, Qiuqiang Kong, Junyan Jiang: A unified model for zero-shot music source separation, transcription and synthesis. ISMIR 2021: 381-388

**Questions:**

Q1: The separation result of ISDM and MSDM in Table 2 suggests a trade-off in terms of generation quality and separation performance. Is it confirmed? If so, why using a unified approach for source separation and generation? I understand on one hand, ISDM helps to show the diffusion method is comparable to the SOTA method. However, since MSDM is always lower than ISDM, I doubt the fundamental assumption in this paper is not valid that it is superior to use joint distribution to model generation and separation at the same time.

Q2: Could you provide some demos for ISDM method (Now or after publication)?

---

> ### Author Response · Authors · 2023-11-20
>
> We thank the reviewer for the positive review and for appreciating novelty, simplicity, clarity and soundness of the proposed method.
>
> **Weaknesses:**
>
> **The idea of using a generative way to model audio and solve multiple audio tasks in one model is not a new idea. For example, we have a unified VAE approach to do source separation and transcription [1]. Of course the current work is novel but it would be better to include those studies and compare the approaches in related work.**
>
> **[1] Liwei Lin, Gus Xia, Qiuqiang Kong, Junyan Jiang: A unified model or zero-shot music source separation, transcription and synthesis. ISMIR 2021: 381-388**
>
> Thank you for pointing this out. We will definitely include and discuss the mentioned paper in our section on generative separation.
>
> **What is the novelty compared to NCSN-BASIS? Any intuition for MSDM Dirac? Is it a marginal improvement from existing methods, or not?**
>
> The main difference between NCSN-BASIS and ISDM is the proposed DIract likelihood, which allows us to consistently improve generation/separation results with respect to the Gaussian one, used in NCSN-BASIS. This better performance may be attributed to Dirac's enforcement of mixture consistency during inference. Moreover, we propose the MSDM model, able to generate consistent mixtures composed of multiple tracks of individual instruments, which is not possible with NCSN-BASIS.
>
> **The introduction of ISDM method is not clear. Is it a baseline method where the assumptions are obviously wrong? Is it another valid approach? Compared to MSDM, is there a trade-off in terms of generation quality and separation performance?**
>
> Thank you for pointing out this lack of clarity in the explanation of the role of ISDM. Essentially the method allows us to demonstrate the effectiveness of generative source separation when combined with Dirac likelihood. It is indeed **not** a valid approach for total and partial generation because the interdependencies between sources are not modeled, and thus it is unable to generate the constituents of a full track, contrary to MSDM (while able to generate a single type of source). This explanation has been added to the section where ISDM is introduced in the revised paper.
>
> **The introduction of correction step is okay. Since it is evaluated in the experiment section, could you explain more in the paper? What is the statement to be expected prior to the experiment regarding correction step?**
>
> Following your suggestion, we moved the section on the sampler details, correction step included, in the main paper. Prior to using correction, we expected the model to have lower performance, given that in literature, predictor-corrector methods usually work better than predictor alone [6]
>
> [6] Yang Song, Jascha Sohl-Dickstein, Diederik P. Kingma, Abhishek Kumar, Stefano Ermon, & Ben Poole.  (2021). Score-Based Generative Modeling through Stochastic Differential Equations.
>
> **Questions**
>
> **Q1: The separation result of ISDM and MSDM in Table 2 suggests a trade-off in terms of generation quality and separation performance. Is it confirmed? If so, why using a unified approach for source separation and generation? I understand on one hand, ISDM helps to show the diffusion method is comparable to the SOTA method. However, since MSDM is always lower than ISDM, I doubt the fundamental assumption in this paper is not valid that it is superior to use joint distribution to model generation and separation at the same time.**
>
> We thank the reviewer for the question. Actually, we do not claim that modeling the joint distribution (MSDM) brings a significant benefit to the separation task, but it enables the generalization of consistent instrumental stems, which is not possible with ISDM.
>
> As for the tradeoff, a direct comparison of the quality of the two models is not possible only through the results contained in Table 2 because the total number of parameters in the two methods is different. Indeed, each model in ISDM uses the same number of parameters as the full MSDM, resulting in the ISDM being able to exploit 4 times parameters to model all 4 instruments. The goal of Table 2 was to show the effectiveness of generative source separation when combined with the proposed Dirac likelihood (in both ISDM and MSDM).
>
> In general, having a single model, like MSDM, for multiple tasks is more versatile, not requiring specialized solutions for different individual tasks, similar to how LLMs can handle different tasks with a single model.
>
> **Q2: Could you provide some demos for ISDM method (Now or after publication)?**
>
> Thanks for your interest, we will certainly provide demos for ISDM separation on the GitHub page after publication.

---

> > ### Comment · Reviewer_yjJy · 2023-11-22
> > **Response to the authors**
> >
> > Thank you for the reply and paper revision. I have no further questions. Your explanation help me understand the work better.

---

### Official Review · Reviewer_K9Yj · 2023-10-30

**Soundness:** 4 excellent
**Presentation:** 3 good
**Contribution:** 3 good
**Rating:** 8
**Confidence:** 3

**Summary:**

This paper proposes a Multi-Source Diffusion Model (MSDM) for performing both music source separation and generation. Modeling multiple sources jointly allows the system to perform source imputation / accompaniment generation by conditioning on a partial mixture during inference. The authors demonstrate the efficacy of MDSM in both source separation and generation tasks, achieving separation performance competitive with state-of-the-art models. Moreover, by modeling sources independently and introducing a novel score function, the authors achieve separation results surpassing the state-of-the-art for certain instrument classes. While generation and imputation abilities appear to be more limited, MSDM demonstrates the strong potential of generative separation systems that model sources jointly.

**Strengths:**

The proposed method is novel and interesting. Given recent advances in generative source separation and the apparent benefits of iterative refinement for separation [1], a diffusion-based approach seems natural.

The proposed method achieves very good separation performance using mostly off-the-shelf components, demonstrating the strength of the general approach. Additional improvements (e.g. weakly-supervised ISDM variant and Dirac score function) are well motivated in the paper.

The problem of source imputation / accompaniment generation is more relevant to many music creation workflows than full mixture generation, and is comparatively under-studied. While the accompaniment generation results here don't necessarily "improve" on those demonstrated in the referenced concurrent work [2] in terms of realism, the proposed method approaches the problem from a different angle (joint source separation and generation, diffusion rather than language modeling) and allows for more fine-grained control using multiple conditioning instrument classes (as opposed to singing only).

[1] Ethan Manilow, Curtis Hawthorne, Cheng-Zhi Anna Huang, Bryan Pardo, and Jesse Engel. Improving
source separation by explicitly modeling dependencies between sources. In ICASSP 2022-2022
IEEE International Conference on Acoustics, Speech and Signal Processing (ICASSP), pp. 291–295.
IEEE, 2022.

[2] Chris Donahue, Antoine Caillon, Adam Roberts, Ethan Manilow, Philippe Esling, Andrea Agostinelli,
Mauro Verzetti, Ian Simon, Olivier Pietquin, Neil Zeghidour, et al. Singsong: Generating musical
accompaniments from singing. arXiv preprint arXiv:2301.12662, 2023.

**Weaknesses:**

I think the paper would benefit from some additional discussion of the computational and data requirements of the proposed method. Presumably separation time is linear in the number of inference steps (plus correction); it would be nice to see this explicitly compared to Demucs with and without Gibbs sampling. Similarly, if the proposed method is more data-hungry than discriminative methods such as Demucs (e.g. if the proposed method is not competitive when trained/evaluated on MUSDB), this might be worth emphasizing further. Parameter counts for each method would also be nice to see.

Based on the listener study and provided listening examples, MSDM struggles to produce coherent and high-quality generations -- even when judged in the context of its synthetic training data. To my ears, it seems like tempo sometimes degrades within the model's 12-second context for unconditional generations, and for imputation generations when strongly metric signals (drums, bass) are not given as conditioning. Overall, the separation results seem much stronger than the generation results.

**Questions:**

The fact that the ISDM method outperforms MSDM on certain sources (Table 2) seems contrary to the intuition that jointly modeling dependencies between sources should improve separation results. Could the authors elaborate on these results, and perhaps conjecture as to why independently modeling sources with ISDM improves (or at least does not substantially deteriorate) performance versus jointly modeling with MSDM?

Given the apparently high data requirements of MSDM, have the authors explored fine-tuning the Slakh2100 model on smaller datasets (e.g. MUSDB)? The authors mention potentially using the outputs of a source separation system to scale up the dataset (akin to SingSong), but fine-tuning also seems like an interesting avenue to explore -- especially given its popularity with diffusion models more generally.

Did the authors conduct any separation or generation experiments with out-of-distribution data?

---

> ### Author Response · Authors · 2023-11-20
>
> We wish to thank the reviewer for the thorough and positive review and for appreciating the novelty and relevance of the idea proposed.
>
> **Weaknesses:**
>
> **I think the paper would benefit from some additional discussion of the computational and data requirements of the proposed method. Presumably, separation time is linear in the number of inference steps (plus correction); it would be nice to see this explicitly compared to Demucs with and without Gibbs sampling. Similarly, if the proposed method is more data-hungry than discriminative methods such as Demucs (e.g. if the proposed method is not competitive when trained/evaluated on MUSDB), this might be worth emphasizing further. Parameter counts for each method would also be nice to see.**
>
> To address requests about inference time and parameter count, we've included a new table, Table 4, in the appendix of the revised paper. This table displays these metrics for the models mentioned in Table 2 (MSDM, ISDM, Demucs and Demucs + Gibbs). Additionally, Table 6 provides metrics for training and evaluation on MusDB.
>
> Regarding parameter count and data efficiency, as expected, our methods require more resources. Demucs is indeed a regressor trained specifically for separation, while MSDM is also generative, and generative models are notoriously parameter and data-hungry (for reference, Mousai is trained on 2500 hours of audio and has 857M parameters, more than double our parameter count).
>
> According to Table 4, inference times for 12 seconds of audio on a single GPU are approximately 0.1s for Demucs, 5s for MSDM, 20s for ISDM, 30 seconds for Demucs + Gibbs (256 steps), and 1 minute for Demucs + Gibbs (512 steps). Demucs + Gibbs (256 steps) was added because 256 is the minimum number of steps that makes the SI-SDRi over all instruments (17.59) greater than the one of ISDM. This makes ISDM time-competitive compared to Demucs + Gibbs (256 and 512 steps).
>
> **Based on the listener study and provided listening examples, MSDM struggles to produce coherent and high-quality generations -- even when judged in the context of its synthetic training data. To my ears, it seems like tempo sometimes degrades within the model's 12-second context for unconditional generations, and for imputation generations when strongly metric signals (drums, bass) are not given as conditioning. Overall, the separation results seem much stronger than the generation results.**
>
> Maintaining coherence for 12 seconds of audio is indeed a challenging task for all generative models. There is extensive literature on how mitigate this issue, such as by employing finer controls like rhythmic structure, as seen in approaches like [5]. With separation, this issue does not arise because long-term coherence is inherently provided by the input mixture.
>
> [5] Shih-Lun Wu, Chris Donahue, Shinji Watanabe, & Nicholas J. Bryan.  (2023). Music ControlNet: Multiple Time-varying Controls for Music Generation.

---

> > ### Author Response · Authors · 2023-11-20
> >
> > **Questions**
> >
> > **The fact that the ISDM method outperforms MSDM on certain sources (Table 2) seems contrary to the intuition that jointly modeling dependencies between sources should improve separation results. Could the authors elaborate on these results, and perhaps conjecture as to why independently modeling sources with ISDM improves (or at least does not substantially deteriorate) performance versus jointly modeling with MSDM?**
> >
> > Unfortunately, a direct comparison of the quality of the two models is not possible only through the results contained in Table 2 because the number of parameters in the two models is different. Indeed, each model in ISDM uses the same number of parameters as MSDM, resulting in the ISDM being able to exploit 4 times parameters to model all 4 instruments. The goal of Table 2 was to show the effectiveness of generative source separation when combined with Dirac likelihood (in both ISDM and MSDM). Nevertheless, we can speculate that information about the joint distribution is redundant for the separation task, as the mixture itself is provided as the conditioning input.
> >
> > **Given the apparently high data requirements of MSDM, have the authors explored fine-tuning the Slakh2100 model on smaller datasets (e.g. MUSDB)? The authors mention potentially using the outputs of a source separation system to scale up the dataset (akin to SingSong), but fine-tuning also seems like an interesting avenue to explore -- especially given its popularity with diffusion models more generally.**
> >
> > In Table 6 in the revised paper, we added also metrics for MSDM trained on Slakh2100 and finetuned on MUSDB on the common Bass and Drums sources. We observe that with respect to the model trained on Slakh and tested on MUSDB, fine-tuning has a very positive effect, however not competitive with Demucs.
> >
> > **Did the authors conduct any separation or generation experiments with out-of-distribution data?**
> >
> > As answered in the first weakness, Table 6 also reports the evaluation for MusDB test set on MSDM trained on Slakh. We can clearly see that MusDB is too out of distribution for the model to generalize.

---

> > > ### Comment · Reviewer_K9Yj · 2023-11-22
> > > **Response to authors**
> > >
> > > I thank the authors for addressing my comments. I think the paper will benefit from the proposed revisions.

---

### Official Review · Reviewer_J4Tn · 2023-11-01

**Soundness:** 4 excellent
**Presentation:** 4 excellent
**Contribution:** 3 good
**Rating:** 8
**Confidence:** 4

**Summary:**

The authors propose to tackle the source separation problem by seeing it as a separate-tracks music generation problem.
They model this distribution using a diffusion model.
This approach allows to tackle more use cases such as "source inputation" (accompaniment generation) using the same model.

**Strengths:**

The paper is well written, well-organized, self-contained and the background and reference section are comprehensive and detailed. The proposed method is sound and the authors obtain positive results. Even if not state of the start (source separation is a highly studied task), this original method shows promising results and has the advantage of being conceptually simpler and more general.

The new proposed modeling method IDSM Dirac seems relevant, clearly improves on previous methods and might be of interest in other application domains.

This method is of course more costly in terms of the amount of data required or in terms of computational resources needed. But this is addressed in the limitations section.

The appendix showcases interesting hyperparameter searches about the MSDM Dirac approach, like the impact on the constrained source.

**Weaknesses:**

Adding details on the correction step could make the paper even more self-contained.

**Questions:**

It would have been great to compare the inference time between the different methods, as the diffusion-based methods are likely to be order of magnitudes bigger.

---

> ### Author Response · Authors · 2023-11-20
>
> First of all, we wish to thank the reviewer for the positive review and for appreciating the novelty, clarity, soundness, and generalizability of the proposed method.
>
>
>
> **Weaknesses:**
>
> **Adding details on the correction step could make the paper even more self-contained.**
>
> We thank the reviewer for pointing out this improvement. In the revised paper, we have moved the section on the sampler to the main paper and clarified some passages. Let us know if this makes it clearer.
>
> **Questions:**
>
> **It would have been great to compare the inference time between the different methods, as the diffusion-based methods are likely to be order of magnitudes bigger.**
>
> We thank the reviewer for the question. We reported a table with the requested inference times in the supplementary material. According to Table 4, inference times for 12 seconds of audio on a single NVIDIA RTX A6000 are approximately 0.1s for Demucs, 5s for MSDM, 20s for ISDM, 30 seconds for Demucs + Gibbs (256 steps) and 1 minute for Demucs + Gibbs (512 steps). Demucs + Gibbs (256 steps) was added because 256 is the minimum number of steps that makes the SI-SDRi over all instruments (17.59) greater than the one of ISDM. While ISDM and MSDM are not time-competitive to Demucs, as the reviewer pointed out, they are more time-efficient compared to Demucs + Gibbs (256 and 512 steps).

---

### Official Review · Reviewer_oyrq · 2023-11-11

**Soundness:** 3 good
**Presentation:** 4 excellent
**Contribution:** 4 excellent
**Rating:** 8
**Confidence:** 3

**Summary:**

My review did not seem to have the right visibility, so I am re-submitting.

The authors present an approach for mixture-based diffusion modeling, complete with a novel inference approach that leverages a posterior based on Dirac delta functions and a Monte Carlo approximation of the Dirac likelihood in place of the mixture density. The authors also demonstrate that this approach can be used for generation. The authors evaluate for stem separation using Slakh2100 against established baselines, and also show quantitative and qualitative human-evaluated results for a generation task, establishing a new task and baseline in the process.

**Strengths:**

- The approach the authors proposed is novel and in fact quite general. Orthogonally, I am interested in understanding the potential of the approach in settings with other mixture-based diffusion.
- The manuscript is compelling to read, well organized, and very clear. The discussion of related work is quite comprehensive.
- Using a waveform as the representation of data generalizes the approach further, opening the door to applicability in other domains.
- The authors provide a strong baseline for accompaniment generation with a data-rich setup (Slakh2100) which will serve to be a solid foundation. If not already planned, I would encourage the authors to release as much as they can with respect to evaluation methodology and reproducible artifacts that others can use to evaluate in a similar manner.

**Weaknesses:**

The results of the paper would be made stronger with more discussion of the computational footprint and details for training and inference. How does the footprint compare to Demucs or other methods? How does does computation scale with the amount of data?

Further, some brief qualitative analysis of the results might be warranted. Do the authors have an explanation for the relative performance on certain stems relative to Demucs? What are the high-level takeaways from the qualitative results in Table 3 beyond those that the reader could intuit or speculate about?

Some feedback on the manuscript:
- Section 2.1, in the last paragraph: should "the minute" read "a minute" denoting duration of the context length?

**Questions:**

- Do the authors have a hypothesis as to why in Table 2 performance on the indicated stem categories (i.e. bass, drums) outperforms Demucs? Is there any intuition as to the variance in results across the techniques?
- Is there an understanding of the approach's data efficiency? How much does Slakh2100 versus MusDB alone? What about compared to other settings in Demucs?

---

> ### Author Response · Authors · 2023-11-20
>
> We would like to express our gratitude to the reviewer for their thorough and positive review, as well as for recognizing the method's clarity, novelty, and generalizability. Regarding its generalizability, the method is indeed inherently adaptable in scenarios where the mixed signal is the sum of sub-signals, for example different components in 3D models.
>
> **Weaknesses discussion:**
>
> **The results of the paper would be made stronger with more discussion of the computational footprint and details for training and inference. How does the footprint compare to Demucs or other methods? How does computation scale with the amount of data?**
>
> We provided a table (Table 4) showing inference times for Demucs, Demucs + Gibbs, ISDM, and MSDM in the appendix of the revised paper. According to Table 4, inference times for 12 seconds of audio on a single GPU are approximately 0.1s for Demucs, 5s for MSDM, 20s for ISDM, 30 seconds for Demucs + Gibbs (256 steps) and 1 minute for Demucs + Gibbs (512 steps), making ISDM time-competitive compared to Demucs + Gibbs (256 and 512 steps). Gibbs (256 steps) was added because 256 is the minimum number of steps that makes the SI-SDRi over all instruments (17.59) greater than the one of ISDM.
>
> As regards training times, our models (MSDM and the four stems of ISDM) on Slakh were trained for around 400 epochs each, employing around 46 minutes per epoch, while MSDM on MusDB was trained for 1700 epochs, each one lasting around 11 minutes. The only comparative training time available is for Demucs on MusDB, because training times for the models trained on Slakh are not reported in [1]. Traning Demucs v2 on MusDB employs 360 epochs, each one lasting around 4 minutes. All the timing is intended on a single GPU, an NVIDIA V100 with 16GB of RAM for Demucs and an NVIDIA Quadro RTX 6000 with 24GB of RAM for our models.
>
> [1] Ethan Manilow, Curtis Hawthorne, Cheng-Zhi Anna Huang, Bryan Pardo, and Jesse Engel. Improving source separation by explicitly modeling dependencies between sources. In ICASSP 2022-2022 IEEE International Conference on Acoustics, Speech and Signal Processing (ICASSP).
>
> **Do the authors have an explanation for the relative performance of certain stems relative to Demucs? What are the high-level takeaways from the qualitative results in Table 3 beyond those that the reader could intuit or speculate about?**
>
> For Demucs, MSDM, and ISDM, separating Guitar and Piano presents a notable challenge due to their similar sound profiles in Slakh2100. This similarity often leads to source spilling between the two stems during separation, potentially harming performance. Interestingly, Gibbs sampling seems to enhance the performance of Demucs + Gibbs in these cases. The performance boost likely stems from Gibbs sampling's property of fixing one source while generating the other.
>
> Looking at Table 3, we notice that increasing the number of conditioning sources (thereby reducing the number of generated sources) consistently improves the FAD score in a near-monotonic manner. This improvement is logical, considering that more conditioning data simplifies the generation task, and more ground truth sources are included in the evaluated mix. However, when generating a single source (Drums) while conditioning on the other three (Bass, Guitar, Piano), we observe a decline in FAD score. This suggests that the strong rhythmic structure provided by the drums makes it easier to generate the other sources, in line with reviewer K9Yj's observations. In future work, we plan to explore the integration of additional conditioning elements, like the controls in the study found at [2], which could complement the conditioning sources in our model.
>
> [2] Shih-Lun Wu, Chris Donahue, Shinji Watanabe, & Nicholas J. Bryan.  (2023). Music ControlNet: Multiple Time-varying Controls for Music Generation.

---

> > ### Author Response · Authors · 2023-11-20
> >
> > **Questions:**
> >
> > **Do the authors have a hypothesis as to why in Table 2 performance on the indicated stem categories (i.e. bass, drums) outperforms Demucs? Is there any intuition as to the variance in results across the techniques?**
> >
> > From Table 2 we can observe that the same model with Gaussian likelihood achieves inferior separation results, which suggests that the effective good performance of ISDM with respect to Demucs and Demucs + Gibbs is more attributable to the Dirac likelihood than to generative-based separation itself. This is likely because the mixture consistency enforced by Dirac during inference assists in the separation task. However, this advantage becomes less noticeable for the guitar and piano stems, as their similar sound profiles make them difficult to separate using a generative-based approach, which explains the inferior performance of ISDM and MSDM reported in Table 2. Note that without the 512 Gibbs steps, even a regressor like Demucs, trained exclusively on the separation task, has a significantly inferior performance on the piano with respect to the other stems.
> >
> > **Is there an understanding of the approach's data efficiency? How much does Slakh2100 versus MusDB alone? What about compared to other settings in Demucs?**
> >
> > In Table 6 in the appendix of the revised paper, we added SISDR metrics for a model trained on MusdBD alone. As we can see, the performances are not comparable with Demucs, being the latter specifically trained for the separation task. Generative properties are notoriously data-hungry (for example, Mousai [3] has been trained on more than 2500 hours of audio).
> >
> > As for the Demucs settings, we are unsure of what the reviewer is asking; could the reviewer please clarify it?
> >
> > [3] Schneider, F., Jin, Z., & Scholkopf, B. (2023). Moûsai: Text-to-Music Generation with Long-Context Latent Diffusion*. arXiv preprint arXiv:2301.11757*.

---

### Author Response · Authors · 2023-11-20
**Summary of changes to revised paper**

We wish to thank the reviewers for their positive feedback and recognition of our method's novelty, clarity, conceptual simplicity, and generalizability to other domains.

Based on the reviewers' recommendations, we made the following changes to the main paper:

1. We included a table containing inference times and the number of parameters in the revised paper (Table 4).
2. We included a table in Appendix F of the revised paper (Table 6) where we compare the performance metrics of Demucs against MSDM either tested, finetuned, or trained on MusDB.
3. Moved the section on the sampler (Section 4.2.4) to the main revised paper. There we specify and explain further the functioning of the correction step.
4. Other minor changes.

These changes are highlighted in blue in the revised paper.

---

> ### Comment · Reviewer_yjJy · 2023-11-21
> **Is the revised paper available now?**
>
> Dear authors and reviewers,
>
> I'm trying to post a private message to you.
> It seems the pdf on the paper page is still the original version and there is no files in the "revisions" link. A reminder to the authors of possible uploading failure. Or maybe I did not find the way?

---

> ### Author Response · Authors · 2023-11-21
>
> Dear reviewer,
>
> Thank you for notifying this! Now it should be avalilable. Thank you again.

---

### Meta-Review · Area_Chair_Qzyq · 2023-12-07

**Metareview:**

The authors present a diffusion based model for source separation and music synthesis, and is one of the first approaches that can perform both generation and separation in a unified framework. In the process, they also propose a novel inference approach based on Dirac likelihoods. All the reviewers highlighted the novelty of the approach, along with its generality / applicability to related tasks. As was pointed out by the reviewers, while the approaches presented do not always significantly outperform contemporary methods on all tasks considered, it does provide a compelling alternative to tasks like source imputation / music generation.

General feedback during the review is about providing details on computational complexity and comparisons with approaches like Demucs. The authors included an analysis on inference times in the Appendix as part of the rebuttal. There were a few discussions about the intuitions as to why the method works compared to other approaches in certain settings, which the authors addressed satisfactorily.

In terms of weaknesses, there were also some interesting questions around performance on out-of-domain data, and addressing data requirements of the method. Indeed, these are important challenges that need to be addressed in future work.

**Justification For Why Not Higher Score:**

N/A

**Justification For Why Not Lower Score:**

All reviewers thought the paper was really novel, providing a new way to address generation and separation in a single diffusion framework. The new inference strategy using Dirac likelihoods is also an important contribution. Given the novelty of the presented approaches, the recommendation is to Accept with Oral. There are some limitations, and the quality is not a significant leap compared to other approaches from the literature on all tasks. This could be a reason to Accept with Spotlight. But the merits outweigh them.

---

### Decision · Program_Chairs · 2024-01-16

Accept (oral)